# Inducible Expression of Several *Drosophila melanogaster* Genes Encoding Juvenile Hormone Binding Proteins by a Plant Diterpene Secondary Metabolite, Methyl Lucidone

**DOI:** 10.3390/insects13050420

**Published:** 2022-04-29

**Authors:** Sang-Woon Shin, Jun-Hyoung Jeon, Ji-Ae Kim, Doo-Sang Park, Young-Joo Shin, Hyun-Woo Oh

**Affiliations:** 1Core Facility Management Center, Korea Research Institute of Bioscience and Biotechnology, Daejeon 34141, Korea; jiaekim@kribb.re.kr; 2Biological Resource Center, Korea Research Institute of Bioscience and Biotechnology, Jeongeup 56212, Korea; wjs258@kribb.re.kr (J.-H.J.); dspark@kribb.re.kr (D.-S.P.); 3Department of Radiation Oncology, Sanggye Paik Hospital, Inje University, Seoul 01757, Korea; shinyj@paik.ac.kr

**Keywords:** juvenile hormone binding protein (JHBP), juvenile hormone disruptor (JHD), methyl lucidone, plant diterpene secondary metabolite (PDSM)

## Abstract

**Simple Summary:**

Multiple genes encoding juvenile hormone binding proteins are present in all insect species. However, the variety of juvenile hormones is limited in insects. This suggests other roles for juvenile hormone binding proteins in addition to their role as juvenile hormone transporters. Here, we show that seven *Drosophila melanogaster* juvenile hormone binding protein genes are inducible by methyl lucidone, a plant diterpene secondary metabolite, which functions as a juvenile hormone disruptor both in vitro and in vivo. This suggests that the diversity of juvenile hormone binding protein genes may be related to the presence of diverse plant diterpene secondary metabolites.

**Abstract:**

Juvenile hormones prevent molting and metamorphosis in the juvenile stages of insects. There are multiple genes encoding a conserved juvenile hormone binding protein (JHBP) domain in a single insect species. Although some JHBPs have been reported to serve as carriers to release hormones to target tissues, the molecular functions of the other members of the diverse JHBP family of proteins remain unclear. We characterized 16 *JHBP* genes with conserved JHBP domains in *Drosophila melanogaster*. Among them, seven *JHBP* genes were induced by feeding the flies with methyl lucidone, a plant diterpene secondary metabolite (PDSM). Induction was also observed upon feeding the juvenile hormone (JH) analog methoprene. Considering that methyl lucidone and methoprene perform opposite functions in JH-mediated regulation, specifically the heterodimeric binding between a JH receptor (JHR) and steroid receptor coactivator (SRC), the induction of these seven *JHBP* genes is independent of JH-mediated regulation by the JHR/SRC heterodimer. Tissue-specific gene expression profiling through the FlyAtlas 2 database indicated that some *JHBP* genes are mainly enriched in insect guts and rectal pads, indicating their possible role during food uptake. Hence, we propose that JHBPs are induced by PDSMs and respond to toxic plant molecules ingested during feeding.

## 1. Introduction

Some juvenile hormone binding proteins (JHBPs), which are insect-specific, have been shown to bind to the juvenile hormones (JH) and have been proposed to function as JH transporters to target tissues and cells [1,2,3,4]. Moreover, JHBP protects JH against degradation by JH esterases [5]. JH biosynthesis occurs in the *Corpora allata* gland, which is located beneath the brains of insects [6]. JHs are then transported to target tissues and cells to mediate JH-dependent regulation, which affects the development [7], reproduction [8], diapause [9], and polyphenism [10], virtually targeting all tissues. To transport JHs and protect them against degradation, JHBPs must be produced by head tissues or secretory tissues into the hemocoel cavity and be present in circulation.

Although JHBPs have been implicated as JH transporters and protect JHs against degradation, the presence of multiple JHBPs suggests roles other than JH transport. Here, we characterized genes with a conserved JHBP domain in *Drosophila melanogaster* genome release 6 (dm6 by the Berkeley Drosophila Genome Project). In contrast, previous reports indicated the presence of a single juvenile hormone, JHIII, in *D. melanogaster* [11]. More recently, JHIII bisepoxide (JHB_3_) was also detected in whole-body extracts of adult *D. melanogaster* [12]. The presence of excessive copies of *JHBP* in the *D. melanogaster* genome compared to the number of JHs may indicate the presence of JHBPs with novel functions other than transporting JHs. Indeed, two proteins in the JHBP family in *D. melanogaster*, Daywake (dyw, *JHBP8*) and Takeout (to, *JHBP9*), have different roles from those of JH transporters. Dyw functions in neurons as a day-specific anti-siesta gene with little effect on sleep levels during the night or in the absence of light [13]. The protein takeout is implicated in the circadian control of feeding behavior [14,15] and affects male courtship behavior [16].

Null mutants of the methoprene-tolerant (Met) gene exhibit strong resistance to both JH and the JH analog (JHA) methoprene [17,18]. Therefore, Met has been characterized as the JH receptor (JHR) in *D*. *melanogaster*. The germ cell-expressed (GCE) gene, which is a *Met* paralog, was identified as a redundant JHR in *D. melanogaster* [17,19,20,21]. Other insects, except the lower dipterans, possess a single functional *Met* gene [22]. Insect JHRs bind to JH with a high affinity and activate the transcription of JH-dependent genes [23,24,25]. JHR, as with the other members of the bHLH-PAS family of transcription factors, requires other bHLH-PAS proteins for optimal functioning [26]. The bHLH-PAS domain-containing steroid receptor coactivator (SRC; i.e., beta FTZ-F1 interacting SRC, FISC in *Aedes aegypti*, or Taiman in *D*. *melanogaster*) interacts with JHR as a heterodimeric partner during JH-dependent gene regulation in *A*. *aegypti* [27], *Tribolium castaneum* [28], and the silkworm *Bombyx mori* [24].

Recently, we found that many plant diterpene secondary metabolites (PDSMs) interfere with the JH-mediated binding of JHR and SRC. Among these JH disruptors (JHDs), methyl lucidone (ML) strongly blocked the larval development of *D*. *melanogaster*, thereby preventing the formation of pupae and adult fruit flies [29]. In this paper, we report that seven of the 16 *D. melanogaster JHBP* genes were strongly (*p* < 0.01) induced by ML. We also found that four of the seven inducible *JHBP* genes were expressed mainly in the rectal pad, which suggests that their role may be linked to ingestion, as well as JH transport. Combined with the results revealing that the *JHBP* genes are inducible by the feeding of ML, we propose that the induction by PDSM activates JHBPs to interact with and possibly counteract potentially harmful plant molecules ingested during feeding.

## 2. Materials and Methods

### 2.1. Chemicals

The plant diterpene, methyl lucidone, was isolated from *Lindera erythrocarpa* as described previously [30]. Methoprene was purchased from Sigma-Aldrich (St. Louis, MO, USA), and each reagent was prepared as a stock solution in ethanol.

### 2.2. Insect Rearing and Feeding

Twenty male and 20 female adult flies were added to individual vials, each containing a 3-g artificial diet mixed with either 0.25% ML (*w/v*), 0.05% methoprene (*w/v*), or 0.25% ethanol (*w/v*), as a control. After 2 d of oviposition, the adult flies were removed from the vials, and the eggs laid were allowed to develop. At this sublethal concentration, ML treatment did not affect their development into adults [29]. After five to six days, the second instar larvae were collected from each vial, and the total RNA was isolated. The RNA was subjected to quantitative polymerase chain reaction (qPCR) analysis (conducted independently in triplicates). To obtain pupae and adult fruit flies, we continued to incubate the eggs until pupariation or eclosion, respectively. The pupae and adult flies (female or male) were collected from each vial, and the isolated total RNA was subjected to qPCR analysis (conducted independently in duplicates).

### 2.3. RNA Extraction, Primers, and qPCR Analysis

An RNeasy kit (Qiagen, Hilden, Germany) was used to extract total RNA from second instar larvae, pupae, adult females, and adult males that were fed ethanol (control)-, ML-, and methoprene-supplemented diets. cDNAs were synthesized for qPCR using a Tetro cDNA Synthesis Kit (Bioline, London, UK) and 1 μg RNA, as estimated using a NanoDrop ND-1000 Spectrophotometer (Thermo Scientific, Waltham, MA, USA).

Gene-specific primer pairs listed below were designed for the 16 *JHBP* genes using Primer 3.

*JHBP1*-F: 5′-GCTGAAGAACATGGAGGCCTTC-3′

*JHBP1*-R: 5′-CCAGAACCAAGCTGAGAGCATC-3′

*JHBP2*-F: 5′-GACAACATCGCCAATGGCAAC-3′

*JHBP2*-R: 5′- CGGAATCTTGGCGAATATGCG-3′

*JHBP3*-F: 5′-AAGTTTTTGTAAACCAATTG-3′

*JHBP3*-R: 5′-CGCAAGTTGTGAACACTGCTC-3′

*JHBP4*-F: 5′-CGTGCAAACGCTCTAATCCG-3′

*JHBP4*-R: 5′-GTTGTACTTCGCCCGGATCCTA-3′

*JHBP5*-F: 5′-GAATCGGACTACAGCATTAAGG-3′

*JHBP5*-R: 5′-CTTGACCTTCACAGTGTTGATC-3′

*JHBP6*-F: 5′-GTGACTAACCCGCTTAGCAGC-3′

*JHBP6*-R: 5′-GATTTGGTTTCGGATCGAGCG-3′

*JHBP7*-F: 5′-GTGACTAACCCGCTTAGCAGC-3′

*JHBP7*-R: 5′-GCGACACGTTGATGATGCAG-3′

*JHBP8*-F: 5′-CAAAGGTTTGGATTACCGCCG-3′

*JHBP8*-R: 5′-GCAGCTCCTTGTTGTCGTTG-3′

*JHBP9*-F: 5′-CCTTCTCACTCGTTGGACCC-3′

*JHBP9*-R: 5′-CGGTAACGTCCAGGTAGGTC-3′

*JHBP10*-F: 5′-CAACGGAGGGAACTATACAGGG-3′

*JHBP10*-R: 5′-CGAAGATACCGAGACCCACG-3′

*JHBP11*-F: 5′-GATTACTAATGCTACTCCATGC-3′

*JHBP11*-R: 5′-GCATGGAGTAGCATTAGTAATC-3′

*JHBP12*-F: 5′-CTTTCCCAAGTGCAAACGGG-3′

*JHBP12*-R: 5′-TCATGTCGTGCACCTTCACA-3′

*JHBP13*-F: 5′-GTGTCCGTCAAACTCAATGGC-3′

*JHBP13*-R: 5′-CTCCACATCCAAGAGCGTTAC-3 ′

*JHBP14*-F: 5′-GATCCGGTGCTGAACGATGTC-3′

*JHBP14*-R: 5′-CGGTAGTGCAGCAAAGAAGTC-3′

*JHBP15*-F: 5′-GACAGTTGCCTCCTGAGATCG-3′

*JHBP15*-R: 5′-CCGACTAACGACGACATCCTC-3′

*JHBP16*-F: 5′-CTACTTCACAACAGGTGTGCCC-3′

*JHBP16*-R: 5′-GAACTTGGTCACCTCGGATCG-3′

qPCR was performed using the RealFAST SYBR kit (Geneer, Daejeon, Korea) in 48-well plates on an Eco Real-Time PCR System (Illumina, San Diego, CA, USA). The following two-step thermal cycler program was used for all runs: 95 °C for 3 min; 40 cycles of 95 °C for 5 s and 60 °C for 20 s; and a final melting curve analysis spanning 95 °C for 15 s, 55 °C for 15 s, and 95 °C for 15 s. The Eco Manager Software (Illumina) was used to validate the amplification efficiency and specificity. RNA extraction and qPCR analysis were performed in triplicate, and the average values of each sample were compared.

### 2.4. Gene Annotation and Subsequent Data Analyses

The 16 *JHBP* genes harboring a JHBP domain(s) were annotated from the D. melanogaster reference genome dm6. For the annotation, the following protein databases were used: Pfam [31], PF06585 and SMART [32], and SM000700. Almost all 16 JHBP family proteins contain only one JHBP domain, except for JHBP10 (two JHBP domains annotated from Pfam). The majority of JHBP proteins contained a signal peptide, as detected by the SignalP v4.0 program [33], except for JHBP4 (CG10264) and JHBP10 (CG33680), which contained a transmembrane helix region (detected by TMHMM v2.0 [34]), instead. For tissue-specific expression of *JHBP* genes, genome-wide tissue-specific gene expression profiles available in the FlyAtlas 2 database [35] were used. FlyAtlas 2 is a repository of tissue-specific gene expression based on genome-wide RNA-seq analyses of *D. melanogaster* genes in adult males, females, and larvae.

## 3. Results

### 3.1. Six among 16 Genes Harboring a JHBP Domain Are Inducible by ML at the Larval Stage

We observed the induction of six of the 16 *JHBP* genes upon ML treatment in all three biological replicates (Figure 1). The inducible expression of these six genes was also observed upon methoprene treatment (Figure 1). The expression profiles of the other 10 *JHBP* genes are shown in Appendix A.

### 3.2. Regulation of Four Larval-Inducible JHBP Genes at the Pupal Stage by ML and Methoprene

We tested the expression of six larval-inducible *JHBP* genes at the pupal stage after feeding ML or methoprene (Figure 2). Four of the six *D. melanogaster* larval-inducible *JHBP* genes were relatively abundantly expressed at the pupal stage, and their expression was significantly affected by both ML and methoprene. Among these, two *JHBP* genes (*JHBP2* and *JHBP4*) were significantly induced by ML, whereas the expression of the other two genes (*JHBP11* and *JHBP12*) was almost completely repressed in the pupal stage (Figure 2). The expression levels of the other two larval-inducible *JHBP* genes (*JHBP1* and *JHBP16*) and the other ten *JHBP* genes at the pupal stage were not affected by ML and/or methoprene (Appendix A).

### 3.3. Stage-Specific Expression of 10 JHBP Genes

Next, we tested the stage-specificity of all 16 *JHBP* genes (Figure 3), which were not inducible upon ML treatment at the larval stage. Some genes showed relatively strong expression at the adult stage: *JHBP9* and *JHBP10* in the adult females, *JHBP5* in the adult males, and *JHBP3*, *JHBP14*, and *JHBP15* in both the adult females and males. *JHBP7* was highly expressed during the pupal stage. *JHBP6*, *JHBP8*, and *JHBP15* were expressed at relatively low abundances at all stages tested.

*JHBP8* encodes Daywake (dyw), which functions as a day-specific anti-siesta gene in neurons [13]. The strongly inducible expression of the *JHBP8* gene by ML at adult stages (Figure 4) suggests an additional role for Daywake in the interaction with ML. We did not observe significant induction of *JHBP8* by methoprene.

### 3.4. Tissue Specificity of JHBP Genes

Although we did not directly test the tissue-specific expression of *JHBP* genes, we utilized genome-wide tissue-specific gene expression profiles available in the FlyAtlas 2 database [35]. FlyAtlas 2 is a repository of tissue-specific gene expression based on genome-wide RNA-seq analyses of *D. melanogaster* genes in adult males, females, and larvae. Figure 5 shows the tissue specificity of *JHBP1*. The FlyAtlas 2 results of other *JHBP* genes are shown in Appendix A.

## 4. Discussion

In this study, we characterized seven *JHBP* genes inducible by ML: six *JHBP* genes inducible at the larval stage and *JHBP8* inducible at the adult stage (summarized in Table 1). The induction of six larval-inducible *JHBP* genes is independent of the JH-mediated regulation by the JHR/SRC heterodimer since the induction was also observed upon feeding the JHA methoprene. Methoprene activates JH-mediated regulation, specifically facilitating the heterodimeric binding between the JHR and SRC. JH-mediated heterodimeric binding between JHR and SRC activates a downstream cascade of JH-dependent genes [27]. In contrast, ML hinders the JH-mediated formation of JHR/SRC heterodimers. If the expression of these inducible *JHBP* genes were regulated by the JHR/SRC heterodimer, methoprene and ML would have an opposite function on the expression of the *JHBP* genes.

The induction of six larval-inducible *JHBP* genes could be mediated by JH in a JHR/SRC heterodimer-independent manner, such as through the juvenile hormone-activated phospholipase C pathway [36]. However, this pathway is activated by JHIII but not by methoprene, as shown in the paper. Therefore, this pathway may not be involved in the regulation of these *JHBP* genes at the larval stage, which are induced by both methoprene and ML.

Since our results on ML-inducible *JHBP* genes may be independent of JH regulation, we propose that these ML-inducible *JHBP* genes are induced in response to PDSMs. Many plant species contain compounds with JHD activity. In all three plants tested: *L. erythrocarpa*, *Solidago serotina*, and *Pinus densiflora* [30,37], the compounds were shown to be diterpene secondary metabolites. PDSMs disrupt insect development by interfering with JH receptor complex formation. We have previously observed that treatment with PDSMs blocks larval development [29,30,37]. Nonetheless, it is still unclear whether the blocking of larval development by PDSMs depends on JH-mediated regulation. PDSMs are present in plants at high concentrations; therefore, we previously observed that the same percentage of PDSMs as the crude plant extracts resulted in identical phenotypes blocking larval development [29,30,37]. The uptake of PDSMs during the digestion of plants would be harmful to insects irrespective of their function as a JHD in a JH-dependent manner or as agents blocking larval development in a JH-independent manner. In either scenario, PDSMs exert toxic effects on larval development during plant feeding and need to be detoxified by insect systems. JHBPs may act as response molecules against these PDSMs. Interestingly, two *JHBP* genes, ML-inducible *JHBP4* and non-inducible *JHBP10*, harbor a transmembrane helix region at the N-terminus instead of a signal peptide (Table 1), indicating that these JHBPs would be unable to function as JHBP transporters. Another remarkable is the finding that four of the seven ML-inducible *JHBP* genes are mainly expressed in the adult rectal pad (Figure 4 and Appendix A), which is involved in water and solute uptake during food ingestion.

Four ML-inducible *JHBP* genes at the larval stage, *JHBP1*, *JHBP2*, *JHBP11*, and *JHBP12*, were relatively abundantly expressed at the pupal stage, and their expression was significantly affected by both ML and methoprene. The expression of two *JHBP* genes, *JHBP1* and *JHBP2*, were induced by ML at both the larval and pupal stages. In contrast, *JHBP11* and *JHBP12* gene expressions were repressed at the pupal stage, whereas they remained inducible at the larval stage. Interestingly, one of the major tissues expressing *JHBP11* and *JHBP12* includes the rectal pad. If *JHBP11* and *JHBP12* expression are linked to PDSM ingestion, their differential expression between the larval and pupal stages may be associated with their role during the uptake and detoxification of PDSMs. Since fruit flies do not ingest food during pupal development, the genes induced at the larval stage would become unnecessary and would need to be shut down at the pupal stage.

In this study, we elucidated the expression profiles of 16 *JHBP* genes and linked them to defensive roles against the PDSM, methyl lucidone. Further studies on the direct interaction between JHBPs and PDSM are necessary to prove this unequivocally.

## Figures and Tables

**Figure 1 insects-13-00420-f001:**
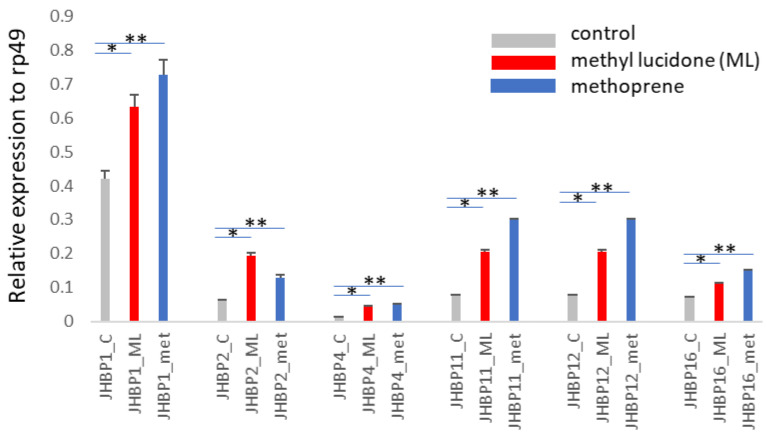
Six *D. melanogaster JHBP* genes are inducible by ML and methoprene at the second instar larval stage. The gene expression was evaluated by qPCR with gene-specific primers. All three biologically independent replicates showed the induction of these six *JHBP* genes, and a representative result is presented. Each result shows the average value of three replicates during the RNA isolation and qPCR steps, and the error bars indicate standard deviation. Statistical significance was determined by a *t*-test. *, *p* < 0.01 between control and ML; **, *p* < 0.01 between control and methoprene.

**Figure 2 insects-13-00420-f002:**
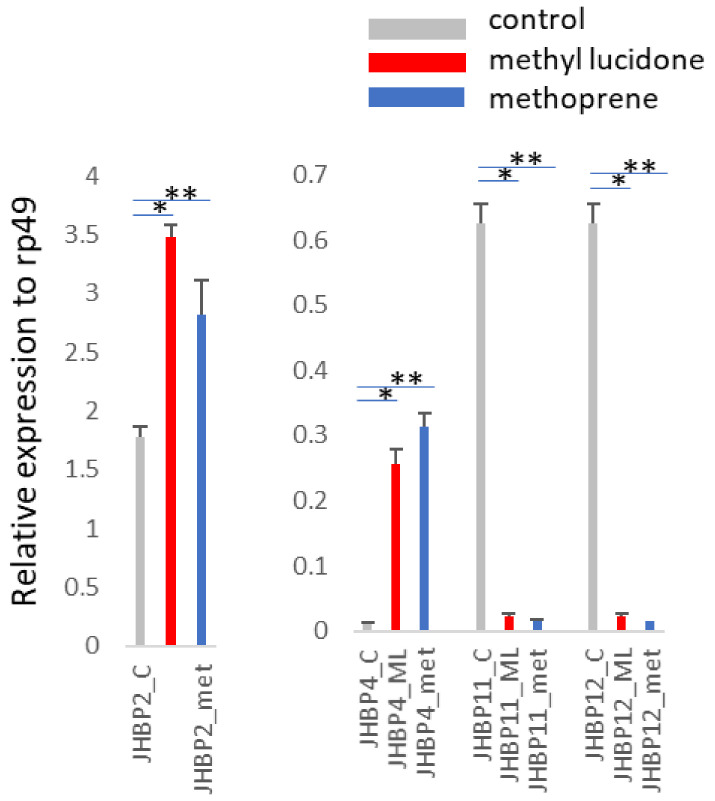
Inducible or repressed expression of four of the six *D. melanogaster* larval-inducible *JHBP* genes at the pupal stage upon ML and methoprene treatment. The gene expression levels were evaluated by qPCR with gene-specific primers. We have shown two figures with different scales of the *Y*-axis to clarify the expression level of each *JHBP* gene: in the left figure, *JHBP2* with relatively high abundance is depicted; on the right, the expression of the three low-abundant *JHBP* genes is shown. Two independent biological replicates showed the same results, and a representative result is presented. Each result shows the average value of three replicates during RNA isolation and qPCR steps, and the error bars indicate standard deviation. Statistical significance was determined by a *t*-test. *, *p* < 0.01 between control and ML; **, *p* < 0.01 between control and methoprene.

**Figure 3 insects-13-00420-f003:**
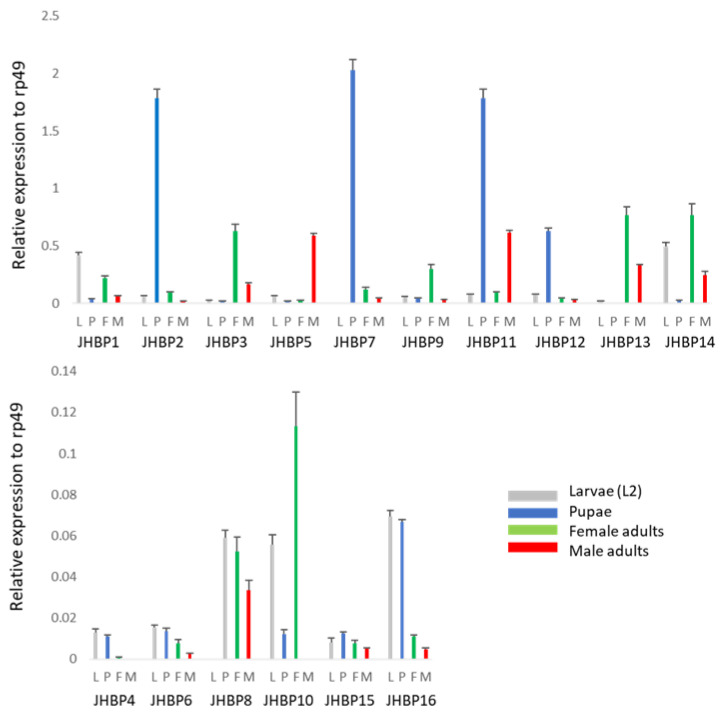
Stage-specificity of *D. melanogaster JHBP* genes. The second instar larvae (L), pupae (P), eclosed adult females (F), and eclosed adult males (M) were collected from fruit flies grown on a normal diet and were subjected to RNA isolation. The gene expression was evaluated by real-time PCR with gene-specific primers. We have shown two figures with different scales of the *Y*-axis to clarify the expression level of each *JHBP* gene: the upper figure depicts 10 *JHBP* genes with a relatively high abundance; the lower figure shows the six *JHBP* genes with a relatively low abundance. Two independent biological experiments showed similar results; representative results are shown in the figure. Each result shows the average value of three replicates during RNA isolation and qPCR steps, and the error bars indicate standard deviation.

**Figure 4 insects-13-00420-f004:**
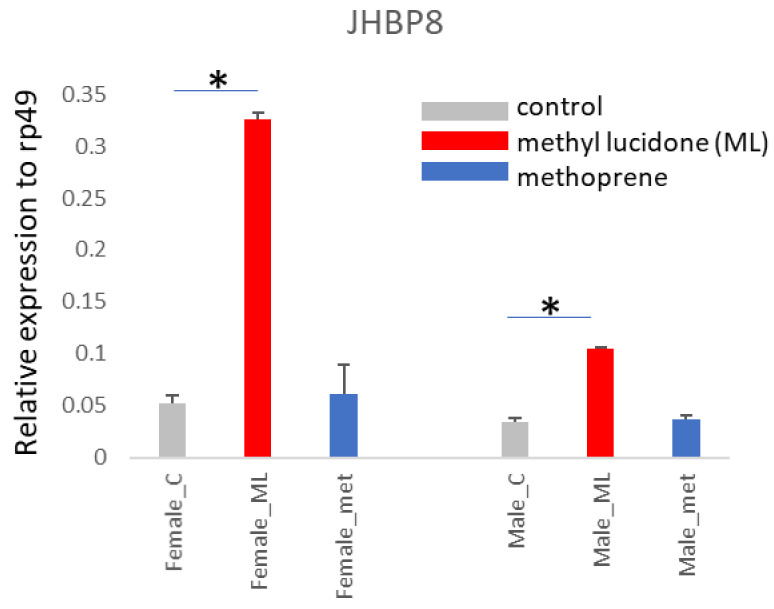
The *JHBP8* gene, which encodes Daywake, is inducible by ML in adult insects. Gene expression was evaluated by real-time PCR with gene-specific primers. A representative result from two biological replicates is depicted. Each result shows the average value of three replicates during RNA isolation and qPCR steps; error bars indicate standard deviation. Statistical significance was determined by a *t*-test. *, *p* < 0.01.

**Figure 5 insects-13-00420-f005:**
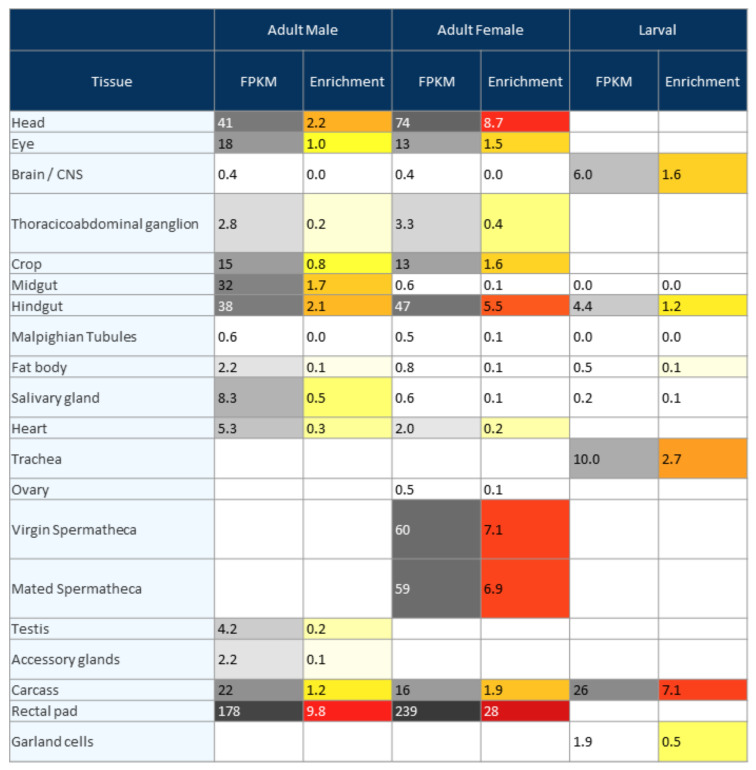
Tissue specificity of *JHBP1* (CG1124) retrieved from the FlyAtlas 2 database.

**Table 1 insects-13-00420-t001:** Characteristic summary of 16 *JHBP* genes. *, genes harboring the putative transmembrane region instead of the putative signal peptide; **, genes encoding proteins with two JHBP domains. Seven inducible *JHBP* genes are indicated in red.

JHBPs	GeneID	Gene Expression (Inducibility by Methyl Lucidone)	Flyatlas 2 Enrichments	Note
JHBP1	CG1124	Inducible in larval stage (Figure 1)	Larval carcass and adult rectal pad	
JHBP2	CG2016	Inducible in larval and pupal stages (Figure 1 and Figure 2)	Larval trachea and adult head	
JHBP3	CG14661	Mainly expressed in adults, no significant induction (Figure 3)	Female head and carcass	
JHBP4	CG10264	Inducible in larval and pupal stages (Figure 1 and Figure 2)	None	*
JHBP5	CG10407	Mainly expressed in male adults (Figure 3), no significant induction	Male testis	
JHBP6	CG14457	only basal expression in all stages (Figure 3), no significant induction	Larval and adult midgut	
JHBP7	CG11852	Mainly expressed in pupal stage (Figure 3), no significant induction	Larval trachea and hindgut	
JHBP8	CG2650	Mainly expressed and inducible in adults (Figure 4)	Male rectal pad	Daywake
JHBP9	CG11853	Mainly expressed in female adults (Figure 3), no significant induction	Ubiquitous	Takeout
JHBP10	CG33680	Mainly expressed in larvae and female adults (Figure 3), no significant induction	None	*, **
JHBP11	CG17189	Inducible in larval stage, but strongly repressed in pupal stage (Figure 1 and Figure 2)	Female rectal pad	
JHBP12	CG13618	Inducible in larval stage, but strongly repressed in pupal stage (Figure 1 and Figure 2)	Adult crop and rectal pad	
JHBP13	CG5945	Mainly expressed in adults, no significant induction (Figure 3)	Adult head	
JHBP14	CG5867	Mainly expressed in larvae and adults, no significant induction (Figure 3)	Ubiquitous	
JHBP15	CG17279	only basal expression in all stages (Figure 3), no significant induction	Adult head	
JHBP16	CG15497	Inducible in larval stage (Figure 1)	None	

## Data Availability

FlyAtlas 2 (https://flyatlas.gla.ac.uk/FlyAtlas2/index.html?page=gene accessed on 25 March 2022) for *JHBP* gene-specific tissue specificity, SMART for Pfam, SignalP v4.0, and TMHMM v2.0 analysis (http://smart.embl-heidelberg.de/ accessed on 25 March 2022).

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
