# Peer review of "Inducible Expression of Several Drosophila melanogaster Genes Encoding Juvenile Hormone Binding Proteins by a Plant Diterpene Secondary Metabolite, Methyl Lucidone"

_insects, 2022, doi:10.3390/insects13050420_

Round 1
Reviewer 1 Report
Methyl lucidone (ML) is a plant diterpene that acts as a juvenile hormone disruptor (JHD) by interfering with the binding of the coactivator SRC (also known as Taiman) with Met when it binds JH and therefore interferes with JH action. This paper reports that ML induces 7 of the 16 JH binding protein (JHBP) genes found in Drosophila when fed to larvae. Methoprene, a JH analog, also induces the same JHBPs. They also show which are present in the pupae and the adult as well as show the tissue distribution of each of them taken directly from FlyBase2 (JHBP1 in the main text and the remainder in the Supplemental Information). Since most of those that are inducible by ML are found in the gut, especially the rectal pads, they conclude that some of the JHBPs are induced as a detoxifying mechanism.
This study is a follow-up to their earlier study on plant diterpenes and disruption of larval development in Drosophila. The work seems to be carefully done. In some cases as noted below, the data could be more clearly presented. Table 1 is a needed and very useful summary of the data. Clearly further work is needed to show conclusively that these particular JHBPs have detoxification of plant terpenoids as their primary function. This work however can stand alone once revised.
Specific comments:
1) In Figure 2, JHBP4 is clearly not as abundantly expressed as the other 3 JHBPs. In fact, one cannot really see its level. It would be good to add an inset of the values for JHBP4 on an expanded Y-axis so that one can see how much lower the expression really is.
2) In Figure 3, JHBP15 is almost undetectable on the figure scale so as in (1) an inset with an expanded Y-axis would help the readers.
3) lines 209,21, These data should be shown, either in the existing figures in the text or in the supplementary information.
4) line 211: By convention, DYW should be capitalized when one talks about its function as a protein.
5) line 225: In the supplemental Figure S1, the series begins with two pages both labeled JHBP3 although the information on the two sheets is different. I assume that the first one should be JHBP2, but check.
6) Table 1. It would be better to say: “Inducible in” and “expressed in” rather than “inducible at” or “expressed at” and also “mainly” rather than “majorly”.
7) line 247: I don’t think that reference 31 is the correct reference for this statement. You also need a reference for the next sentence. AT this point there is not that much known about the role of the phospholipase C pathway in JH action in the larva so your statement concluding that “this pathway is not involved in the regulation of these JHBP genes at the larval stage” is purely speculative and not warranted based on your data.
8) line 263: detoxified rather than detoxicated
9) lines 278 to 282: At metamorphosis the entire larval gut dies and is replaced by imaginal cells which exist as islands in the larval gut, then in the pupal proliferate to form the adult gut. Therefore, the loss of JHBP 11 and JHBP12 expression in the pupa is likely primarily due to the loss of those parts of the gut that were highly expressing in the larva.
Author Response
We would like to thank the reviewer for their time and effort spent in evaluating our manuscript. We believe that their thoughtful comments have helped improve the manuscript. We have addressed their concerns in detail to the best of our abilities, as presented below.
Specific comments of the reviewer #1:
1) In Figure 2, JHBP4 is clearly not as abundantly expressed as the other 3 JHBPs. In fact, one cannot really see its level. It would be good to add an inset of the values for JHBP4 on an expanded Y-axis so that one can see how much lower the expression really is.
> Instead of adding an inset of the values for JHBP4, we have separated Figure 2 into two with high and low transcript abundance to clarify the expression of each JHBP gene. In addition, “abundant expression” has been changed to “inducible or repressed expression” in line 188.
2) In Figure 3, JHBP15 is almost undetectable on the figure scale so as in (1) an inset with an expanded Y-axis would help the readers.
> Instead of adding an inset of the values for JHBP15, we have separated Figure 3 into two figures with high and low transcript abundance to clarify the expression of each JHBP gene.
3) lines 209-212, these data should be shown, either in the existing figures in the text or in the supplementary information.
> We have presented the methoprene results in Figure 4.
4) line 211: By convention, DYW should be capitalized when one talks about its function as a protein.
> We have fixed the error.
5) line 225: In the supplemental Figure S1, the series begins with two pages both labeled JHBP3, although the information on the two sheets is different. I assume that the first one should be JHBP2, but check.
> Thank you for bringing this discrepancy to our attention. We have fixed it in Figure S3.
6) Table 1. It would be better to say: “Inducible in” and “expressed in” rather than “inducible at” or “expressed at” and also “mainly” rather than “majorly”.
> We have made the changes according to the reviewer’s suggestion.
7) line 247: I don’t think that reference 31 is the correct reference for this statement. You also need a reference for the next sentence. AT this point there is not that much known about the role of the phospholipase C pathway in JH action in the larva so your statement concluding that “this pathway is not involved in the regulation of these JHBP genes at the larval stage” is purely speculative and not warranted based on your data.
> We apologize for the mistake. The reference (31) has been changed to (36) and the text has been changed according to the reviewer’s comment.
8) line 263: detoxified rather than detoxicated
> We have rectified the error.
9) lines 278 to 282: At metamorphosis the entire larval gut dies and is replaced by imaginal cells which exist as islands in the larval gut, then in the pupal proliferate to form the adult gut. Therefore, the loss of JHBP 11 and JHBP12 expression in the pupa is likely primarily due to the loss of those parts of the gut that were highly expressing in the larva.
> The loss of JHBP 11 and JHBP12 expression in the pupa occurred only with the treatment of ML or methoprene. JHBP11 and 12 were strongly expressed in pupae without the treatment (Figure 3).
Reviewer 2 Report
The manuscript "Inducible Expression of Several Drosophila melanogaster 2 Genes Encoding Juvenile Hormone Binding Proteins by a Plant 3 Diterpene Secondary Metabolite, Methyl Lucidone" characterized 16 JHBP genes with conserved JHBP domains in Drosophila melanogaster. Among them, seven JHBP genes were induced by feeding the flies with methyl lucidone, a plant diterpene secondary metabolite (PDSM). Induction was also observed upon feeding the juvenile hormone (JH) analog methoprene. The results suggest that JHBP is induced by PDSMs and responds to toxic plant molecules ingested during feeding. However, there are several issues to be addressed.
Line 78. 'p' should be italic.
Line 98-102. Here should be to put forward the scientific question of this study, rather than give the research results. This section should be moved to the Discussion.
The description of data analysis method is missing in "Materials and Methods".
Line 160-163. This part belongs to " Materials and Methods" and should not appear here.
Figure 1. Only six genes that were significantly up-regulated were shown, and the results of other genes need to be provided in the supplementary. There is the same problem in other Figures.
Figure 3. Statistical significance is not marked in the figure. In addition, t-test is not suitable for analysis here.
Line 193-194. Why only these 10 genes were measured. Does not match the description of Line 193-194 - " Almost none of the 16 JHBP genes from D. melanogaster were inducible by ML at the adult stage (data not shown), except JHBP8".
In addition, data mentioned in the manuscript that are not shown need to be shown in the supplementary.
Author Response
We would like to thank the reviewer for their time and effort in critically evaluating our manuscript. We believe that their thoughtful comments have helped improve the manuscript. We have addressed their concerns in detail to the best of our abilities, presented below.
Specific comments of the reviewer #2:
1) Line 78. 'p' should be italic.
> We have made the necessary changes throughout the manuscript.
2) Line 98-102. Here should be to put forward the scientific question of this study, rather than give the research results. This section should be moved to the Discussion.
> We could not find the research results in lines 98-102. If the reviewer’s comment means lines 77-82, we have minimized the paragraph in order to briefly describe the research results in the introduction section (lines 77-80).
3) The description of data analysis method is missing in "Materials and Methods".
> The sub-section “Gene annotation and subsequent data analyses” has been added in the Materials and Methods section.
4) Line 160-163. This part belongs to " Materials and Methods" and should not appear here.
> We have removed the suggested part from the results section and added the sentence, “At this sublethal concentration, ML treatment did not affect their development into adults [29],” to the Materials and Methods section.
5) Figure 1. Only six genes that were significantly up-regulated were shown, and the results of other genes need to be provided in the supplementary. There is the same problem in other Figures.
> We have supplemented Figures 1 & 2 with Figures S1 & S2 to show the result of other genes. In addition, we have depicted all 16 JHBP genes in Figure 3.
6) Figure 3. Statistical significance is not marked in the figure. In addition, t-test is not suitable for analysis here.
> Apologies for the mistake. We have removed the erroneous sentence.
7) Line 193-194. Why only these 10 genes were measured. Does not match the description of Line 193-194 - " Almost none of the 16 JHBP genes from D. melanogaster were inducible by ML at the adult stage (data not shown), except JHBP8".
> We completely agree with the reviewer’s comment that the expression profiles of all 16 genes have to be shown when we say that “Almost none of the 16 JHBP genes from D. melanogaster were inducible by ML at the adult stage (data not shown), except JHBP8.” We apologize for the error in the description caused by a lack of conclusive data to support our argument. We were unable to repeat the experiments except those for JHBP8. Therefore, we have removed the sentence, “Almost none of the 16 JHBP genes from D. melanogaster were inducible by ML at the adult stage (data not shown), except JHBP8.”
8) In addition, data mentioned in the manuscript that are not shown need to be shown in the supplementary.
> We have ensured that no data are mentioned in the manuscript without corresponding figures, supplementary or otherwise. The data not accompanied by figures have been removed from the texts.
Reviewer 3 Report
In this study, the authors determined the transcriptional responses of the 16 JHBP genes to methyl lucidone, a plant diterpene secondary metabolite, and JH analog methoprene in Drosophila melanogaster. The results showed that six genes of the 16 JHBPs were upregulated by both methyl lucidone and methoprene. The methods are detailed, the results are presented clearly, and the discussion is fine.
However, this work is too preliminary to be published, especially considering the powerful insect model used. The authors raised an interesting viewpoint that some JHBPs may serve against plants, but it is hard to be convinced based on only these qPCR results. To some extent, I prefer to think of it as overstate. Many genetic tools, such as RNAi lines, are available in the VDRC and Bloomington stock centers. Unfortunately, the authors did not validate the potential physiological and developmental functions of any JHBPs. Actually, such descriptive work is not even acceptable in non-model insects currently, let alone in Drosophila melanogaster.
Author Response
We would like to thank the reviewer for their time and effort spent in evaluating our manuscript. However, we disagree with this reviewer's critics.
The reviewer criticized that “the viewpoint that some JHNPs may serve against plants is hard to be convinced based on only these qPCR results”. Not only the qPCR results, we also showed genomic annotations of 16 JHBP genes and Flyatlas 2-based tissue-specificity of those 16 genes. We empathized that the higher copy numbers of JHBP genes and their tissue-specificity were key points of our hypothesis that some JHBPs may serve against plants. To improve our hypothesis, we showed that seven JHBP genes were inducible by feeding of a plant diterpene, methyl lucidone, by qPCR analyses. We don’t understand why this kind of genomic approaches can be ignored as incomplete works in the post-genomic era. In addition, we don’t know yet which kinds of genetic flies could be applied to our research following the reviewer's suggestion to improve our paper. Still, we don’t know the mechanism governing JHBP actions against plants. In addition, we have no idea which JHBP mutants will have a mutant phenotype when considering that 16 JHBP genes are present in the genome of Drosophila melanogaster. We think we need more knowledges about the interaction between JHBPs and plant diterpenes. And then we will proceed to solve the mechanism and/or JHBP function against plants through mutant analyses lately.
Round 2
Reviewer 2 Report
Figure 2 needs to be modified. You need to combine two separate figures into a complete one. Changes can be made with reference to the method of figure 5 (Guo et al., 2018). In addition, there are three treatments including control, ML and metagene, so it is not appropriate to use t-test analysis. Figure 1S and 2S also needs to be changed.
Guo H, Huang C, Jiang L, et al. Transcriptome analysis of the response of silkworm to drastic changes in ambient temperature. Applied microbiology and biotechnology, 2018, 102(23): 10161-10170.
Author Response
Thanks for your kind review and spending your time for the improvement of our manuscript.
Specific comments of the reviewer #2:
1) Figure 2 needs to be modified. You need to combine two separate figures into a complete one. Changes can be made with reference to the method of figure 5 (Guo et al., 2018). Figure 1S and 2S also needs to be changed.
> Figure 2 was originally one figure in first submission, but separated to two figures following first reviewer’s comment. Figure S1 & S2 are the same cases.
In Figure 2, JHBP4 is clearly not as abundantly expressed as the other 3 JHBPs. In fact, one cannot really see its level. It would be good to add an inset of the values for JHBP4 on an expanded Y-axis so that one can see how much lower the expression really is. à Instead of adding an inert of the values for JHBP4, we separated Figure 2 to two figures with high and low transcript abundance in order to clarify the expression of each JHBP gene.
2) In addition, there are three treatments including control, ML and metagene, so it is not appropriate to use t-test analysis.
> We are sorry for the confusion. To clarify, we used two different symbols indicating “Statistical significance was determined by a t-test. *, p < 0.01 between control and ML; **, p < 0.01 between control and methoprene.”
Reviewer 3 Report
To me, the present form is still unacceptable. First, feeding experiments and qPCR analyses indeed provide interesting results but no gene function validation. The authors can at least test this by using many RNAi lines available in well-known fly stock centers, but they did not. Second, although they reanalyzed some data from the published database, the results only have a minimal role in supporting their hypothesis and can at best be considered supplementary data.
Author Response
We would like to thank the reviewer for their time and effort spent in evaluating our manuscript. We also respect the reviewer’s opinion that the paper is still unacceptable. However, the reviewer did not answer to our questions about the reviewer’s critics. Why did the reviewer think that our genome-wide approaches are unacceptable? What kind of Drosophila RNAi lines can be used for improving our researches? We have to know the answers to prepare the next submission, if our paper is not accepted in the journal Insects. There are thousands of Drosophila RNAi lines in fly stock centers. Still, we don’t know the mechanism governing JHBP actions against plants. In addition, we have no idea which JHBP mutants will have a mutant phenotype when considering that 16 JHBP genes are present in the genome of Drosophila melanogaster. If we proceed without further knowledges about the regulation mechanisms and if we simply use single JHBP mutant, we are pretty sure we doomed to fail.